# Effects of Hormones on Breast Development and Breast Cancer Risk in Transgender Women

**DOI:** 10.3390/cancers15010245

**Published:** 2022-12-30

**Authors:** Martine Berliere, Maximilienne Coche, Camille Lacroix, Julia Riggi, Maude Coyette, Julien Coulie, Christine Galant, Latifa Fellah, Isabelle Leconte, Dominique Maiter, Francois P. Duhoux, Aline François

**Affiliations:** 1Breast Clinic, King Albert II Cancer Institute, Cliniques Universitaires Saint-Luc, Avenue Hippocrate, 10, 1200 Brussels, Belgium; 2Department of Gynecology, King Albert II Cancer Institute, Cliniques Universitaires Saint-Luc, Avenue Hippocrate, 10, 1200 Brussels, Belgium; 3Department of Plastic Surgery, King Albert II Cancer Institute, Cliniques Universitaires Saint-Luc, Avenue Hippocrate, 10, 1200 Brussels, Belgium; 4Department of Pathology, King Albert II Cancer Institute, Cliniques Universitaires Saint-Luc, Avenue Hippocrate, 10, 1200 Brussels, Belgium; 5Department of Radiology, Cliniques Universitaires Saint-Luc, Avenue Hippocrate, 10, 1200 Brussels, Belgium; 6Department of Endocrinology and Nutrition, Cliniques Universitaires Saint-Luc, Avenue Hippocrate, 10, 1200 Brussels, Belgium; 7Department of Medical Oncology, King Albert II Cancer Institute, Cliniques Universitaires Saint-Luc, Avenue Hippocrate, 10, 1200 Brussels, Belgium

**Keywords:** breast development, breast cancer risk, transgender women, exogenous hormones, heterogeneity of hormonal treatments

## Abstract

**Simple Summary:**

Transgender women experience gender dysphoria due to a gender assignment at birth that is incongruent with their gender identity. Transgender people undergo different surgical procedures and receive sex steroids hormones to reduce psychological distress and to induce and maintain desired physical changes. A search of the existing literature dedicated to hormone regimens used for treating male to female patients, their impact on breast tissue (tissue development, incidence and type of breast lesions observed) and breast cancer risk provided the available information for this review. An evaluation of breast cancer risk is complicated because of the heterogeneity of administered treatments and a lack of long-term follow-up studies.

**Abstract:**

Transgender women experience gender dysphoria due to a gender assignment at birth that is incongruent with their gender identity. Transgender people undergo different surgical procedures and receive sex steroids hormones to reduce psychological distress and to induce and maintain desired physical changes. These persons on feminizing hormones represent a unique population to study the hormonal effects on breast development, to evaluate the risk of breast cancer and perhaps to better understand the precise role played by different hormonal components. In MTF (male to female) patients, hormonal treatment usually consists of antiandrogens and estrogens. Exogenous hormones induce breast development with the formation of ducts and lobules and an increase in the deposition of fat. A search of the existing literature dedicated to hormone regimens for MTF patients, their impact on breast tissue (incidence and type of breast lesions) and breast cancer risk provided the available information for this review. The evaluation of breast cancer risk is currently complicated by the heterogeneity of administered treatments and a lack of long-term follow-up in the great majority of studies. Large studies with longer follow-up are required to better evaluate the breast cancer risk and to understand the precise mechanisms on breast development of each exogenous hormone.

## 1. Introduction

Transgender people experience gender dysphoria due to a gender assignment at birth that is incongruent with their gender identity [1,2,3,4]. The estimated proportion of transgender adults in the United States and in Europe has increased from 2007 to 2020 and is still increasing [5]. Data from a registry in the Netherlands show a current incidence of 1.8 to 2% of the general population [6,7]. This review focuses on transgender women, also called transwomen or male-to-female women [8,9]. The precise number of transgender women estimated in a population depends on the chosen definition, and unfortunately, no universal consensus exists. In the Netherlands, transwomen are defined as patients who received female hormones and who underwent a transgender surgery, and a prevalence of one in 11,900 women is reported [7]. In the USA, the definition includes women who received endocrine therapy and/or underwent a transgender surgery, and the reported prevalence is slightly higher: 3.29 women in 10,000 people [10,11]. In a review of 29 studies, the prevalence of transgender women ranged from 5–20 to 521 per 100,000 people, depending on the criteria that were used: endocrine treatment for the studies reporting a lower prevalence or self-identification for the studies with a higher prevalence [12,13].

The female transgender landscape is heterogeneous in terms of populations and particularly complex in terms of treatments and follow-up [11,13,14]. The first aim of this review is to focus on breast development in cis- and transgender patients and to try to compare them in terms of hormonal influences. Hormonal influences on breast development are reviewed [15]. The second aim is to summarize the hormonal regimens used in the main cohort studies of transwomen, and the more recent international guidelines and recommendations concerning these treatments [12,16,17,18]. The impacts of endocrine treatments on benign breast lesions and breast cancer risk in transwomen are discussed [19,20]. The effects of gender-affirming hormonal treatment on the risk of hormone-dependent malignancies (including breast cancer) seem to be largely unexplored [21]. Finally, this review analyzes the existing guidelines of breast screening and breast follow-up in male-to-female patients [6,22,23].

## 2. Breast Development

### 2.1. In Cisgender Women

The mammary gland develops through several distinct stages [24,25,26].

In humans, the earliest breast development occurs prenatally, is hormone independent and does not differ between sexes [15,24]. Morphologically mammary glands are formed by different types of cells: the epithelial cells, which elaborate the ductal network, and the stromal cells, which support the mammary gland. After birth, mammary development is arrested until puberty, when hormonal stimulation triggers further differentiation: duct elongation, dichotomous and lateral branching, terminal duct lobular unit formation and stromal expansion are observed in female adolescents, while the male breast remains quiescent but capable of further development under certain circumstances such as relative estrogen dominance [24,27,28,29,30,31,32]. In females, the onset of puberty promotes different events that transform the rudimentary mammary structure into the mature mammary gland [33]. Estrogens in association with growth hormone (GH) and insulin growth factor-1 (IGF-1), through the activation of estrogen receptor alpha (Erα), cause the growth and transformation of the tubules into the mature ductal system of the breast [15,25,34,35,36,37,38].

Progesterone, in conjunction with GH and IGF1, also affects the development of the breast during puberty and thereafter [34,39,40]. To a lesser extent than estrogens, progesterone also contributes to ductal development and lobulo-alveolar formation [40,41]. During pregnancy, pronounced breast growth and maturation occur in the preparation of lactation and breastfeeding. The estrogen and progesterone levels increase dramatically and cause the secretion of high levels of prolactin [42].

The development and functional differentiation of the mammary gland represents a fantastic and highly complex morphogenetic process involving interactions between key hormones and multiple growth factors [29,43,44] (Figure 1 illustrates breast development in ciswomen and -men).

### 2.2. In Transgender Women

Transgender women also go through different stages of breast development once endocrine therapy with estrogen is started. The complex hormonal network that generates the development of the mammary gland in cisgender women is gradually being elucidated [29]. On the contrary, the situation in transgender women is still mysterious and remains largely unexplored. The interactions between estrogenic treatments and residual endogenous androgen secretion on mammary gland development are not detailed and explained in the literature. Breast development remains a key marker of feminization and occurs predominantly during the first 6 months of gender-affirming hormone therapy. The maximum development may be expected after 2 years of treatment [19,45,46]. Unfortunately, data concerning natural history and the effects of different cross-sex hormone therapies on breast development in transwomen are very scarce and based on low-quality studies. In this context, the role of progestogens on breast development remains uncertain and controversial. Wierckx and colleagues concluded that no argument supported or refuted any impact of progestogens on breast development [47]. On the contrary, there exist prior published data on the potential benefits of progesterone on breast development [48]. Up to 2/3 of transwomen are unsatisfied with their breast development and apply for breast augmentation surgery [49,50]. The optimization of cross-sex hormone therapy in transgender women requires prospective randomized trials to evaluate the role of estrogens and estrogens plus progesterone on breast development [34,43]. The follow-up period, according to the results observed in the study of de Blok, needs to be extended because the authors believe that innovative combinations of gender-affirming treatment could further increase breast size and, in this way, reduce the number of surgical procedures [46].

## 3. Hormone Therapy in Transwomen

The hormonal regimens used for transgender women are not standardized across the world. Guidelines arising from different societies (Endocrine society, WPATH—World Professional Association for Transgender Health) have been published [12,18,51,52].

Hormone replacement therapy has become a cornerstone for the management of gender dysphoria in transgender patients. In transwomen, the major aim of hormone therapy is to better align their physical and psychological features with a more feminine expression phenotype [18]. Sex hormones affect the brain by changing mood and have an effect on libido [13,17,52,53,54].

Supplementation with estrogens lowers testosterone concentrations because of negative feedback on the hypothalamic-pituitary gonadal axis. Conjugated estrogens and synthetic estrogens are not recommended because it is not possible to monitor their blood concentration and these drugs increase the risk of thromboembolism [18,45,55]. After the initiation of estrogen therapy, testosterone concentrations decrease to low ranges for biological males (200–300 ng/dL), but normal testosterone levels for natal females are around 75 ng/dL. The recommended estrogen doses for transwomen are higher than the doses given to women receiving replacement therapy. Natural estrogens should be preferred, and prospective studies with a long follow-up are absolutely necessary to evaluate long-term toxicity and the risks of breast and other cancers [20,56,57].

Many patients will require the addition of an antiandrogen medication to inhibit testosterone production or to block the androgen receptor [58,59,60].

Different androgen receptor blockers can be used. Flutamide, which is administered in patients with prostate cancer, can be used at doses of 50–75 mg/day and is known to generate gynecomastia development. Spironolactone blocks androgens from binding to their receptors and increases both total and free estrogen levels, resulting in antiandrogenic properties. The drug is used at a dose of 100 to 200 mg/day with a maximum of 400 mg/day. Cyproterone acetate was used in MTF patients at doses of 50–100 mg/day, has stronger anti-androgenic effects than spironolactone, but can induce severe side effects such as depression and meningioma, so it is no longer recommended in transwomen [51].

The place of progesterone remains controversial: a possible role to increase breast growth has not been demonstrated and the association of progesterone with estrogens increases the risk of cardiovascular diseases.

GnRH (gonadotropin-releasing hormone) agonists are commonly prescribed in adolescents to block pubertal development [54,61].

The precise concentration that results in adequate feminization with the lowest risk of complications is not currently known [2,62,63].

Physical changes observed include softening of the skin, mood changes, decrease in libido and erections, fat redistribution at the hip, and growth of breast tissue [45]. Up to 2/3 of transwomen are unsatisfied with their breast development and require breast augmentation surgery [64].

The main side effects of taking estrogens are

-An increased thromboembolic risk [55];-Liver disorders;-Lipid abnormalities;-Depression and suicidal risk, which are high among transgender patients [17,52,65,66];-Breast cancer risk [6,20,23,67].

The most commonly used drugs are presented in Table 1.

## 4. Benign Breast Lesions in Transwomen

Benign breast lesions observed in male-to-female patients are essentially reported in case reports, except for data published in the Amsterdam Cohort [7]. Therefore, currently, it seems impossible to prospectively evaluate their incidence. The panel of benign breast lesions observed comprises fibroadenomas, cysts, gynecomastia, benign phyllodes tumors and radial scars [68,69,70]. In the Amsterdam Cohort, many benign breast lesions observed are related to breast surgery and represent complications of breast implants. With the modest data of the literature, it is really difficult to establish risk factors and a time frame for the appearance of lesions after initiation of hormonal treatment. Most often, the reason for consultation that leads to the realization of breast investigations is the self-palpation of a nodule.

Figure 2, Figure 3, Figure 4, Figure 5, Figure 6 and Figure 7 describe a panel of different lesions observed in a patient treated for 2 years by natural estrogens and antiandrogens.

## 5. Risk of Breast Cancer: What Do We Know?

The long-term effects of pharmaceuticals used for feminization in transwomen have not yet been sufficiently studied [71,72].

There is a certain parallelism between substitution treatments for cisgender women and feminization hormone treatments for transwomen. Nevertheless, some differences can be noted. Substitution treatments already have a long history [72,73,74,75,76]. Following the results of various randomized and cohort studies, the most famous of which is the WHI (Women’s Health Initiative) study, and in order to reduce the carcinogenic risk of estrogens and progestogens, these treatments have evolved towards the use of natural compounds at low doses and for a limited period of time [71,74,76]. The results of more recent studies than WHI, and iterative analyses and modeling have suggested that the effect of hormones is to promote the growth of occult tumors. Despite the results of more recent studies, the debate is not closed and the question of the exact impact of hormone replacement therapy on breast cancer risk remains a controversial subject [73,76,77,78,79,80]. The Lancet meta-analysis published in 2019 is complex and includes a large number of studies, totaling more than 100,000 women. What this study shows is that taking estrogens alone for more than five years increases the risk of breast cancer, however not as much as taking combined hormone therapy with progestogens. Moreover, women should be counselled that other factors such as alcohol consumption and obesity have in fact a greater effect on breast cancer risk than hormone replacement therapy [81].

On the contrary, for endocrine treatments in transgender women, information is lacking, as there are no randomized studies. Some cohort studies are available but long-term follow-up is currently lacking. In Anglo-Saxon countries, conjugated estrogens are still regularly used, whereas European countries prefer natural compounds [56]. To obtain an antiandrogenic effect, Europeans use spironolactone, which is less potent than antiandrogens such as cyproterone acetate or flutamide but has a better safety profile [56].

Concerning the risk of breast cancer induced by feminization treatments, it should be noted that the doses used in transgender patients are higher than the doses used as replacement therapy in cisgender women [56].

The issue of treatment duration has only been raised in a few publications and should be the subject of future studies.

In the nationwide cohort study performed in the Netherlands, 2260 transwomen received gender-affirming hormone treatment. Fifteen cases of invasive breast cancer were identified (median duration of hormone treatment 18 years, range 7–37 years); this was 46-fold higher than in cisgender men (SIR 46.7, 95% CI 27.2 to 75.4) but lower than in cisgender women (0.3, 95% CI 0.2 to 0.4) [7,67,82]. The median age at diagnosis was 50 (interquartile range 43–55), compared with a median of 61 years among the general female Dutch population. Most tumors were of ductal origin, and estrogen and progesterone receptors were positive. In this study, the authors observed that the risk of breast cancer increased in a relatively short time, suggesting a rapid development of breast tumors [7,67,68,82,83,84].

Comparison between gender-affirming hormone therapy (GAHT) for transwomen and hormone replacement therapy (HRT) for cisgender women highlights the following differences:-GAHT is sometimes administered before surgical removal of birth-sex gonads, so the patient may have elevated serum levels of both masculinizing and feminizing hormones [54,84].-Schedules of GAHT can vary widely by individual patient, with differences in the levels of hormones. These differences complicate the studies on the role of GAHT on breast cancer risk [20,82,85].-As a general rule, the duration of GAHT in transwomen will far exceed the recommended duration of HRT in ciswomen, and this duration criteria needs to be taken into account and analyzed.

The transgender community is a particularly fragile community that is often discriminated against and does not always have access to quality health care [84,85,86]. As a result, particularly vulnerable patients too often resort to buying drugs on the black market [87,88]. This situation increases the risk of developing various pathologies, including the risk of breast cancer, which will not be properly screened [53,65,89,90,91,92,93].

## 6. Transgender Screening Recommendations for Breast Cancer

There are currently no international established cancer screening guidelines for transgender patients at any point in their transition process [6,13,22,23].

Breast cancer screening guidelines in cisgender women are an evolving area of medicine [94,95,96]. Recent data now recommend personalized breast cancer screening [97,98]. Among transgender women, we need to face the poor understanding of the effect of GAHT and the lack of reliable epidemiologic data [99]. Some questions remain unresolved: At which age is it recommended to begin breast cancer screening? How should the number of years of hormone exposure be integrated [94]?

Screening should be based on shared decision making, and patients and providers may want to start screening at an earlier age in patients with a significant family history [21,94].

That is the reason Maglione et al. recommend screening based on risk factors (such as family history, Klinefelter syndrome, *BRCA1*/*2* mutations, etc.) [23].

On the contrary, Gooren et al. do not recommend screening for transwomen [100,101]. According to their team, the annual rate of breast cancer in transwomen does not seem to be higher than the annual rate of breast cancer in men, which is discordant with other previously cited data [20].

Brown et al. recommend similar screening to that in ciswomen: a mammography every two years between the ages of 50 and 69 [82].

Therefore, the systematic review conducted by Hartley et al. could not make a universal recommendation due to the large number of case reports, the lack of meta-analysis, the significant variation in hormone treatment, the wide variability in treatment uptake and the lack of long-term follow-up [102].

More recently, a cohort study was conducted in the Netherlands by the Medical University of Amsterdam in response to the increasing number of transwomen.

As a result of this study, it is recommended that transwomen undergoing hormonal treatment for at least 5 years be followed up in a similar way to ciswomen [7].

This recommendation is also applied in other countries. For example, in the Belgian hospital UZ Gent, which specializes in the treatment of transgender people, it is recommended to screen transwomen who have been on hormonal treatment for more than 5 years. This recommendation is also followed in Canada [103]. (Recommendations for breast screening in transwomen are shown in Table 2.)

## 7. Conclusions

This review highlights the difficulties of establishing and measuring breast cancer risk among transgender women. It confirms the need to register as many transgender patients as possible and to carry out cohort studies with precise data on hormone treatments and long-term follow-up [16,84,104]. Systematic mammographic screening and radiological follow-up are still too often lacking in the management of transgender women undergoing feminizing hormones. Epidemiological data collected on the age and incidence of breast cancer in transgender women, and the extrapolation of the data to cisgender women are sufficient elements to recommend systematic screening and regular follow-up [23,82]. Other difficulties are due to the particular fragility of the community of transpeople [52,65,66,92,93,105]. Anxiety, depression and suicidal ideation have been well documented in adolescents and adults with gender dysphoria. Social rejection, stigma, discrimination and low access to health care contribute to the increased risk of different health problems in this community [84,86,87,92,105,106,107].

## Figures and Tables

**Figure 1 cancers-15-00245-f001:**
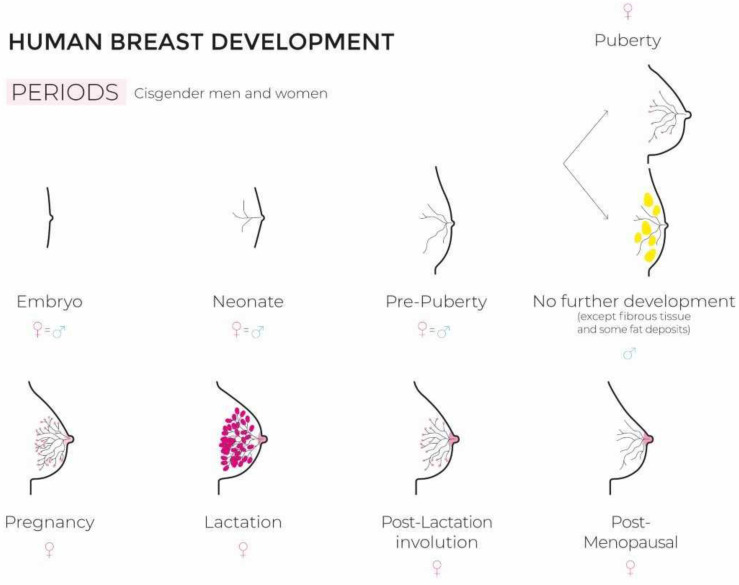
Human breast development (figure created by Florence Boulvain).

**Figure 2 cancers-15-00245-f002:**
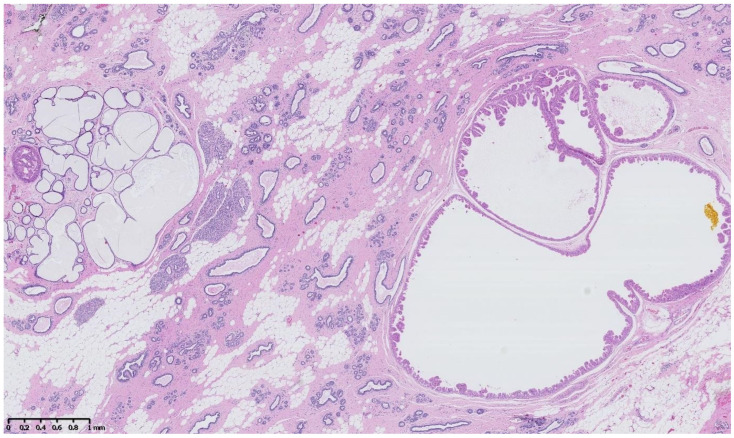
Hematoxylin and eosin staining: the surrounding breast parenchyma shows fibro-cystic changes with apocrine metaplasia, cystic ducts and usual duct hyperplasia.

**Figure 3 cancers-15-00245-f003:**
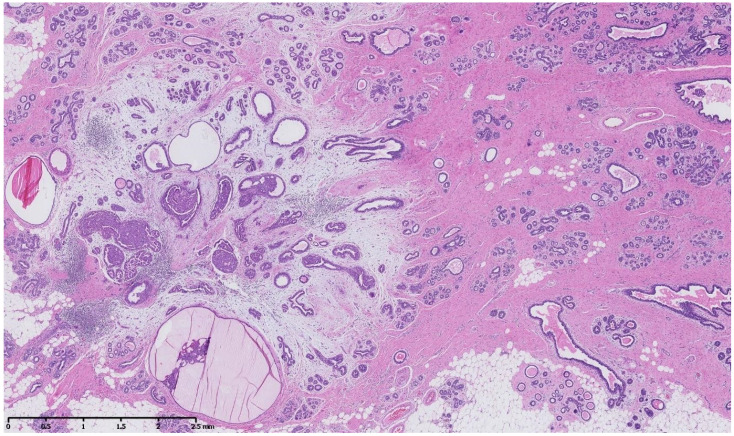
Hematoxylin and eosin staining: at low magnification, the breast parenchyma shows a fibrous stroma containing an irregularly shaped lesion with a fibro-myxoïd dense stroma and inflammatory changes containing cystic ducts and hyperplastic ducts.

**Figure 4 cancers-15-00245-f004:**
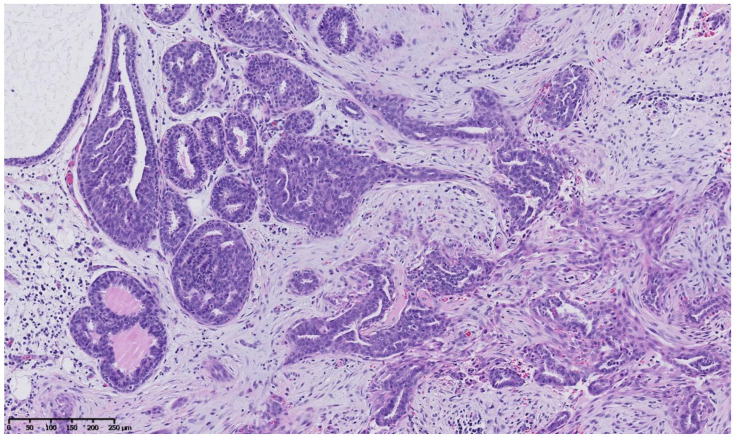
Hematoxylin and eosin staining: at mid magnification, the ducts vary in size, and some look irregularly shaped and pseudo-invasive in the retracted fibrous stroma, whereas others are enlarged and contain an intra-ductal epithelial proliferation.

**Figure 5 cancers-15-00245-f005:**
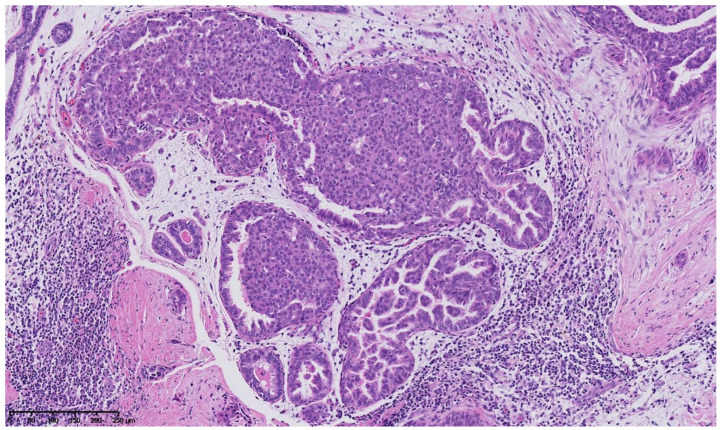
Hematoxylin and eosin staining: at higher magnification, some intraepithelial proliferations are formed by stubby micropapillae with a heterogeneous nuclear population showing no significant cellular atypia, and some are of cribriform architecture with slit-like fenestrations and nuclear streaming, showing no significant cellular atypia either.

**Figure 6 cancers-15-00245-f006:**
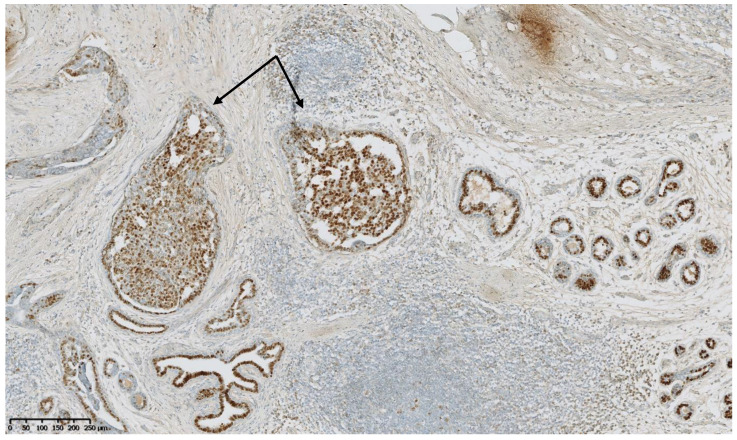
Estrogen receptors immunostaining: the hyperplastic ducts (arrows) show a heterogeneous expression of estrogen receptors, as do the normal ducts and lobules in the vicinity.

**Figure 7 cancers-15-00245-f007:**
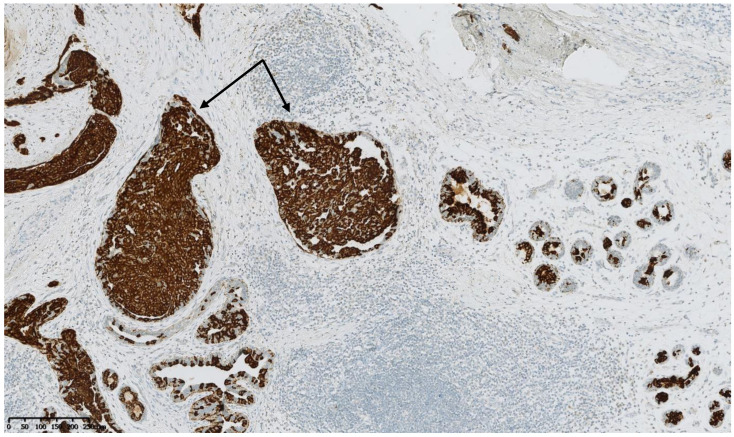
Cytokeratin 5 immunostaining: the intraductal proliferation in the hyperplastic ducts (arrows) still show the presence of myoepithelial cells, as do the normal ducts and lobules in the vicinity.

**Table 1 cancers-15-00245-t001:** Drugs and dosages of HRT (hormone replacement therapy) compounds in cisgender women and GAHT (gender-affirming hormone therapy) in transgender women.

Drugs	Cisgender Women	Transgender Women
Androgens lowering drugs		
Cyproterone acetate	Low doses of cyproterone acetate in some hormonal associations (no longer recommended)	50–100 mg/daily Widely used in the past (T’Sjoen—Patel) Decreased use because of side effects: - Meningioma - Increased risk of severe depression - Liver failure
Spironolactone (T’Sjoen—Patel)	/	100–200 mg orally dailyMax 400 mg
Peripheral androgen receptors blockers (Finasteride)	/	1.5 mg orally daily
GnRH analogs gonadotropin-Releasing hormone)	/	3.75 mg IM (intramuscular) or SC (subcutaneous)/month
Estrogens In the past: oral conjugated estrogens	0.625 mg daily	2.5–7.5 mg daily
Currently: Estradiol valerate		
- Oral	0.5–1 mg/daily	4 mg daily (2–6 mg)
- IM	10–20 mg IM/4 weeks	20 mg IM every 2 weeks
- Transdermal	0.0375–0.05 mg daily (patch changed weekly)	Patch 0.1–0.5 mg/3 days
- Gel spray	Daily gel (0.8–1.5 mg daily)	Daily gel (0.8–3 mg daily) 2 times/day

**Table 2 cancers-15-00245-t002:** Recommendations for screening and follow-up.

Authors	Screening YES/NO	Frequency/Beginning
Maglione et al., 2014 [23]	Yes	Based on risk factors
Goren et al., 2011 [98]	No	/
Brown et al., 2015 [67,82]	Yes	Mammography every 2 years from 50 years old
Hartley et al., 2018 [102]	No recommendation	No recommendation
De Blok et al., 2019 [6]	Yes	Every 2 years = similar as ciswomen
T’Sjoen, 2019 [45]	Yes	After 5 years of hormonal treatment
Canadian Society of Cancer, 2022 Chen [103]	Yes	After 5 years of hormonal treatment

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
