# Peer review of "Effects of Hormones on Breast Development and Breast Cancer Risk in Transgender Women"

_cancers, 2022, doi:10.3390/cancers15010245_

Round 1

Reviewer 1 Report

Effects of hormones on breast development and breast cancer risk in transgender women

The authors present a review on existing literature dedicated to hormone regimens used for treating male to female patients, their impact on breast tissue (tissue development, incidence and type of breast lesions observed) and breast cancer risk. The first aim was to focus on breast development in cis- and transgender patients and try to compare them in terms of hormonal influences on breast development. The second aim was to summarize the hormonal regimens used in the main cohort studies of trans women and the more recent international guidelines and recommendations concerning these treatments. The third aim was to discuss the impact of endocrine treatments on benign breast lesions and breast cancer risk in trans women. Finally, the authors analyzed the existing guidelines of breast screening and breast follow-up in male to female patients.

General comment.

The title of the paper suggests that some findings regarding the use of hormones on breast development and the risk of breast cancer in transgender women will be presented. However, the authors intend to present a review of the literature, and regarding  the effect of hormones use on the risk of breast cancer in transgender women, the authors basically refer to the nationwide cohort study performed in the Netherlands.  

Regarding the literature review relative to the breast development and the use of hormones, the authors did not present the methodology they used to find and select the relevant papers, leaving out important contributions. For example, in rows 212-214 the authors stated that “Despite the results of more recent studies, the debate is not closed and the question of the exact impact of hormone replacement therapy on breast cancer risk remains a controversial subject [72,75,76–79].” They did not include a meta-analysis, published in The Lancet 2019. In that study, it is concluded that “for women of average weight in developed countries, 5 years of MHT, starting at age 50 years, would increase breast cancer incidence at ages 50–69 years.

The paper is presented as a general description of some subjects regarding the use of hormones in transgender woman, that may be useful as an introduction for students or professionals with no experience in the subject. However, the paper lacks of more insightful information regarding the hormone action and differences with cis woman.

Reviewer 2 Report

This review summaries the progress in the field of hormone treatment on transgender women in their breast development and breast cancer risk. It is a very pleasant reading on a thorough summary of the current status of the fields. The review raises more questions than the questions it answers, probably a right reflection of the field considering the scarcity of studies, obviously partly caused by the scarcity of cases, but also due to relatively lower profile of the hormone therapy on trans-gender women comparing with that on cis-women, also a reflection on the limitation of resources.

A number of conclusions are quite striking including that the majority of transgender women do have have satisfactory outcome on breast development following hormone therapy alone hence requested surgical augmentation, which although is a clinical common occurrence but does raise question of further directions: is surgery the answer for all these women or alternative strategies need to be developed to meet the demands?

The other is that trans women on hormone therapy seemingly have lower risk of breast cancer based on the limited number of studies. This is a relief in that these women are generally on higher dose of hormones but again the long term effect is still uncertain due to the limitation of the studies. This again raises more questions on further management.

The authors have done a rather thorough collection of data in the sparsely populated field, and the publication of such review will certainly raise the profile of the studies and hopefully lead many more to come.

A few minor issues:

In Table 1, the authors referred the hormonal treatment to both cis- and trans-women as 'hormone replacement therapy (HRT)'. Although technically it is correct, the fact that HRT has been so publicised as a treatment for menopausal cis-women, personally I think it is probably better to differentiate the treatment to trans women as 'gender affirming hormone treatment (GAHT)' that the authors have used in other parts of the manuscript.

The table is a bit misaligned in a couple of places including the column 3 for 'cyproterone acetate', and column 1 for 'currently: estradiol valerate' 

Figure 5 and 7 do not have legends

Figure 6 referred 'arrows' in the legend but I can't see any in the picture.
